# Pharmaceutical Applications of Supercritical Fluid Extraction of Emulsions for Micro-/Nanoparticle Formation

**DOI:** 10.3390/pharmaceutics13111928

**Published:** 2021-11-14

**Authors:** Heejun Park, Jeong-Soo Kim, Sebin Kim, Eun-Sol Ha, Min-Soo Kim, Sung-Joo Hwang

**Affiliations:** 1College of Pharmacy, Duksung Women’s University, 33, Samyangro 144-gil, Dobong-gu, Seoul 01369, Korea; heejunpark@duksung.ac.kr (H.P.); bernadette0758@duksung.ac.kr (S.K.); 2Dong-A ST Co. Ltd., 21, Geumhwa-ro 105beon-gil, Giheung-gu, Yongin-si 17073, Korea; js_kim@donga.co.kr; 3College of Pharmacy, Pusan National University, 63 Busandaehak-ro, Geumjeong-gu, Busan 46241, Korea; edel@pusan.ac.kr; 4Yonsei Institute of Pharmaceutical Sciences & College of Pharmacy, Yonsei University, 85 Songdogwahak-ro, Yeonsu-gu, Incheon 21983, Korea

**Keywords:** supercritical fluid, supercritical fluid extraction of emulsions, micro-/nanoparticle, pharmaceutical application

## Abstract

Micro-/nanoparticle formulations containing drugs with or without various biocompatible excipients are widely used in the pharmaceutical field to improve the physicochemical and clinical properties of the final drug product. Among the various micro-/nanoparticle production technologies, emulsion-based particle formation is the most widely used because of its unique advantages such as uniform generation of spherical small particles and higher encapsulation efficiency (EE). For this emulsion-based micro-/nanoparticle technology, one of the most important factors is the extraction efficiency associated with the fast removal of the organic solvent. In consideration of this, a technology called supercritical fluid extraction of emulsions (SFEE) that uses the unique mass transfer mechanism and solvent power of a supercritical fluid (SCF) has been proposed to overcome the shortcomings of several conventional technologies such as solvent evaporation, extraction, and spray drying. This review article presents the main aspects of SFEE technology for the preparation of micro-/nanoparticles by focusing on its pharmaceutical applications, which have been organized and classified according to several types of drug delivery systems and active pharmaceutical ingredients. It was definitely confirmed that SFEE can be applied in a variety of drugs from water-soluble to poorly water-soluble. In addition, it has advantages such as low organic solvent residual, high EE, desirable release control, better particle size control, and agglomeration prevention through efficient and fast solvent removal compared to conventional micro-/nanoparticle technologies. Therefore, this review will be a good resource for determining the applicability of SFEE to obtain better pharmaceutical quality when researchers in related fields want to select a suitable manufacturing process for preparing desired micro-/nanoparticle drug delivery systems containing their active material.

## 1. Introduction

Numerous pharmaceutical formulations have been developed in the form of micro-/nanoparticles that comprise composites or encapsulate structures consisting of drugs with or without excipients, such as coating materials, carriers, or stabilizers. In particular, polymers have been most widely used as materials constituting the inner matrix or outer shell of micro-/nanoparticles [1,2,3,4,5]. It is generally known that particles with diameters ranging between 1 and 250 μm and 10 and 1000 nm correspond to microparticles and nanoparticles, respectively [6,7,8,9]. 

There are three major types of micro-/nanoparticles. The first is a matrix-type particle generally produced by co-precipitation of drugs with ingredients such as polymers, coating additives, or other drugs. In this method, micro-/nanoparticles are typically prepared in a state in which the drug molecules are homogeneously distributed in the matrix (Figure 1a). The second method is to obtain typical core-shell structure particles in the form of drug encapsulations of the drug by forming a film layer of the coating material on the outside of the particles (Figure 1b). In some cases, such matrix and encapsulated structures are formed in the particles simultaneously (Figure 1c) [10]. Micro-/nanoparticles prepared by the above two methods are produced in spherical form; hence, the terms “micro-/nanosphere” and “micro-/nanocapsule” have been used interchangeably with “micro-/nanoparticle” [11,12,13,14]. In addition, a method of impregnating a drug substance into micro-/nanoparticles made only of coating materials can also be used to prepare a matrix-type particle.

Such micro-/nanoparticle formulations can be applied to enhance bioavailability and/or ensure successful commercialized manufacturing via controlled drug release, improve dissolution rate and physicochemical stability, and allow for easier handling of fine powder [15,16,17,18,19,20,21,22]. For these purposes, drying and extraction of emulsions have been widely applied for manufacturing micro-/nanoparticles in the pharmaceutical field. In particular, this emulsion-based micro-/nanoparticle preparation technology has been widely used for drug encapsulation, and there are cases where it has been successfully commercialized for final drug products. This emulsion-based micro-/nanoparticle manufacturing process should be capable of mass production, and products manufactured using it should meet the medicinal product quality criteria. Such critical quality necessitates suitable residual organic solvents, along with a uniform size distribution in the desired particle size range, pore size, and volume. In general, the preparation of micro-/nanoparticles form an emulsion via solvent evaporation (SE) or extraction involves two major steps: (i) emulsion preparation and (ii) solidification via removal of the organic solvent (Figure 2) [23]. After preparation of the emulsion, micro-/nanoparticle fabrication is conventionally performed using three basic techniques: SE and/or extraction, phase separation (coacervation), and spray-drying [23,24]. Most of the recently-suggested solvent removal processes of emulsions have been developed by modifying the above three basic processes. A summary of these major solvent removal techniques is presented in Table 1. Although these technologies have been investigated for 50 years, large variation in results within and between production batches is commonly observed. Thus, there is still a need for developing better solvent-removal methods to produce more uniform and reproducible particles [24]. Recently, the pore-closing method, the thermoreversible-gel method, and microfabrication techniques have been introduced as novel technologies, but they are still under investigation owing to the need for improvement [11]. 

As an alternative technology to overcome these shortcomings, manufacturing methods using supercritical fluids have recently been proposed, and their applications have been studied widely. The use of supercritical fluids as an extraction medium has been intensively researched to efficiently eliminate harmful organic solvent residues in the final micro-/nanoparticles, resulting in a process known as supercritical fluid extraction of emulsions (SFEE). In this review, we will deal with this SFEE technology, with a particular focus on application cases in the pharmaceutical field. The pharmaceutical application cases of SFEE were organized and classified according to several types of drug-delivery systems and active pharmaceutical ingredients. Thus, it is expected that this review will guide the evaluation of the applicability of SFEE to obtain better pharmaceutical quality when researchers in related fields are selecting a suitable manufacturing process for preparing desired micro-/nanoparticle drug delivery systems containing their active material.

## 2. Supercritical Fluid Extraction of Emulsions (SFEE)

### 2.1. Supercritical Fluid (SCF)

A SCF is any substance at a temperature and pressure above its critical point that exhibits both gaseous and liquidus properties. This supercritical state is well explained using the pressure-temperature phase diagram (Figure 3a). Based on this phase diagram, fine control of the pressure or temperature near the critical point can lead to large density changes, thereby altering the mass transfer. SCFs possess high diffusivity and low viscosity. This gas-like property can induce efficient mass transfer in the diffusion mechanism (Figure 3b). However, SCFs also exhibit liquid-like densities that can lead to a relatively higher solvent power. In this respect, SCFs have been used as preferred solvents in various pharmaceutical applications, such as extraction and particle formation processes, as alternatives to harmful organic solvents [37,38,39].

Supercritical carbon dioxide (SC-CO_2_) is the most commonly used SCF in the pharmaceutical field because of its low critical temperature and pressure (T_c_ = 304.2 K and P_c_ = 7.38 MPa). This mild temperature condition is an advantage in the pharmaceutical manufacturing process of heat-labile drugs, especially biomolecules such as peptides and proteins. In addition, CO_2_ is nontoxic, nonflammable, and inexpensive. SC-CO_2_ is a solvent approved generally recognized as safe (GRAS) by the United States Food and Drug Administration (US-FDA) [40,41]. It is also considered a green solvent and natural material withdrawn from the environment. This is because CO_2_ emitted industrially or environmentally is collected and used in industrial processes instead of harmful solvents. In addition, since it is cleanly emitted in the environment after use, it does not generate additional CO_2_ or pollute the environment. In addition, CO_2_ is a safe solvent with a high threshold limit value of 5000 ppm [42]. 

Various micro-/nanoparticle manufacturing technologies using SCF, especially SC-CO_2_, have been studied for approximately 30 years. The rapid expansion of supercritical solutions (RESS) using SCF as a solvent and the supercritical antisolvent (SAS) process using SCF as an antisolvent have been widely applied in the pharmaceutical field. Based on these basic principles and the specific effects of SCF, including plasticizing or atomization, several other modified SCF techniques have been introduced: aerosol solvent extraction system, particles from gas saturated solutions (PGSS), supercritical assisted atomization (SAA), and supercritical fluid-assisted spray-drying (SA-SD) [43,44]. However, these SCF techniques have several main issues: irregular particle morphology, un-uniform size distribution, and difficulty in nanoparticle production due to strong particle agglomeration, which makes them difficult to develop as robust pharmaceutical processes with reproducibility of products. This problem is because the glass transition temperature of polymer decreases due to the SC-CO_2_ dissolved or impregnated in the polymer, which can cause the particles to swell, and this can also increase the interconnection between the particles [44]. 

In contrast to previously reported SCF technologies, the SFEE process has been proposed as a useful technology for the preparation of solvent-free micro-/nanoparticles in aqueous suspensions using SCF as an excellent extracting agent for the organic solvent phase of emulsions [24,44]. Many application studies using SFEE technology have been conducted on chemical drugs, biomolecules, inorganic materials, natural antioxidants, lipids, and polymers for controlled release, passive targeting, enhanced dissolution and bioavailability, and physicochemical stabilization of many formulations administered through various routes. The findings in many application studies have shown that a fast extraction rate in the SFEE process results in the prevention of particle aggregation, leading to a narrow particle size distribution (PSD) (when compared with the SAS process) due to the external continuous aqueous phase being immiscible with, especially, SC-CO_2_ [24].

### 2.2. SFEE Process, Apparatus, and Its Extraction Mechanism

#### 2.2.1. SFEE Process and Apparatus

The SFEE process typically includes emulsion preparation as the first step. A stable emulsion with the desired droplet was first prepared using general emulsion preparation methods and subsequently added to the SFEE. The commonly used emulsion preparation methods for micro-/nanoparticle formation are described below [11,23]. In general, the pharmaceutical additives, such as polymers and/or surfactants, are dissolved in a suitable water-immiscible organic solvent. Then, the active drug is dissolved or dispersed in the organic solvent phase. Finally, this primary solution is mixed with a continuous phase, which is typically immiscible with the primary solution (generally aqueous phase), to form a uniform droplet via emulsification [23,26]. In this emulsion preparation process, water-soluble hydrophilic drugs such as peptides and proteins are dispersed in the oil phase as a dissolved solute in the inner aqueous phase (*W/O/W* or *W/O/O*) or dispersed as an undissolved solid (*S/O/W* or *S/O/O*). In contrast, poorly water-soluble hydrophobic drugs are usually dissolved in an organic solution phase with coating/matrix materials (*O/W* or *O/O*) [27]. General critical material attributes (CMA) in the emulsion preparation step are the physicochemical properties (molecular weight, concentration, etc.) of additives such as polymer, lipid, surfactant, diluent, and stabilizer, and the physicochemical properties of the organic solvent used (density, viscosity, boiling point, polarity, affinity for water, vapor pressure, etc.), their content, and volume ratio of aqueous phase to oil phase. In addition, critical process parameters (CPPs) typically involved in the emulsion preparation step are emulsification apparatus, shear force, the material and size of the filter, the feed rate of each phase, temperature, etc. [45]. 

The prepared emulsion is injected into an extraction apparatus filled with SCF and contacted with SCF for rapid removal of the organic solvent; then, the fabricated solid solute remains in a suspension. The size distribution of the emulsion droplets is a major controlling parameter in the emulsion preparation process. In the SFEE process, the temperature, pressure, and flow rates of SC-CO_2_ and emulsion are typical CPPs for controlling solid particle fabrication [46]. Schematic diagrams of various SFEE apparatuses are shown in Figure 4. For the most basic batch-type SFEE process (b-SFEE), the separately prepared emulsion is first placed into the vessel, and then SCF is injected into the bottom of the vessel to flow through the emulsion for efficient extraction of the organic solvent (Figure 4a). During extraction, the temperature and pressure are maintained at optimized supercritical conditions. The organic solvent extracted into the SCF is withdrawn from the top of the vessel. Finally, after solidification using complete solvent removal via extraction, a formed suspension is generally collected from the bottom of the vessel [1]. 

Chattopadhyay and Gupta proposed a novel silica nanoparticle preparation method using SC-CO_2,_ which simultaneously acts as an antisolvent, extraction agent, and reactant. In this study, a *W/O* microemulsion of an aqueous sodium silicate solution was introduced into a SCF batch reactor filled with SC-CO_2_. Rapid extraction of the organic solvent in the oil phase resulted in aqueous sodium silicate reverse micelles, thereby forming silica nanoparticles [47]. This report could be the starting point of the SFEE technology. In 2004, Perrut et al. patented a SCF technology as a method for obtaining solid particles from W/O emulsions [48]. However, in 2006, Chattopadhyay et al. of Ferro Corporation developed and patented a method and apparatus for both batch and continuous particle production from *O/W* or *W/O/W* emulsions using SCF [49]. The patented technology of b-SFEE and continuous SFEE (c-SFEE) was also made available for the production of 10 nm to 100 μm particles with high efficiency [46]. 

In c-SFEE (Figure 4b), SCF is injected from the bottom of the high-pressure extraction vessel to form a desired supercritical condition, and then the emulsion in the separated container is sprayed from the top of the vessel. At this time, the emulsion and SCF are simultaneously and continuously sprayed into the vessel, and the increased contact area of the sprayed small droplets with SCF can further increase the extraction efficiency when compared with the b-SFEE process. Moreover, the solid micro-/nanoparticles precipitated in the form of an aqueous suspension at the bottom of the vessel can be continuously collected by removal through control valve from the vessel [1]. However, the c-SFEE technology of Ferro Corporation has a major limitation in that it is intrinsically discontinuous and semi-continuous. Indeed, only the extraction process is continuous, but the emulsion manufacturing process is discontinuous. The emulsion must be prepared separately batch to batch. This discontinuous emulsion production between batches may lead to critical problems, such as reduced production yield and poor reproducibility [44]. For the continuous emulsion preparation, in 2008, Chattopadhyay et al. proposed a modified SFEE technology where the emulsion formed through the in-line homogenization device was directly put into the SFEE process [50]. Nevertheless, it still had a limitation in that the plant volume had to be increased for scale-up in commercial production.

To overcome this issue and develop an intrinsically c-SFEE available on a commercial scale with improved production efficiency, in 2011, Della Porta et al. proposed an upgraded c-SFEE process using a tower of countercurrent long columns packed with stainless steel instead of a general vessel with a cavity (c-SFEE-PC), which can enhance the extraction efficiency by maximizing the contact area between the emulsion droplet and SCF (Figure 4c). In addition, the long-packed column can also be composed of several multi-stages in which several cylinders are connected by cross unions. The micro-/nanoparticles suspension obtained was continuously collected from the bottom of the column [44,51]. This promising c-SFEE-PC technology can achieve fine control in micro-/nanoparticle formation to achieve desired particle size with narrow PSD, robust and reproducible results, great uniformity within and between batches, and higher throughput using small plants in only a few minutes [44]. 

**Figure 4 pharmaceutics-13-01928-f004:**
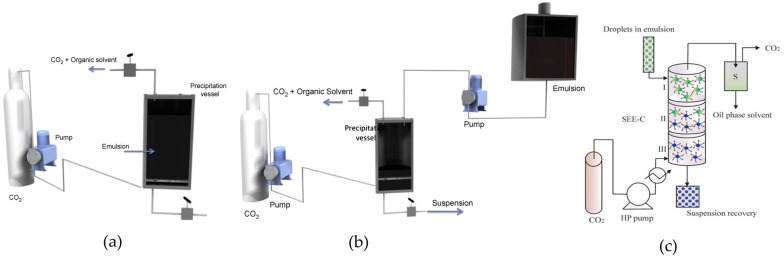
Schematic diagram of SFEE apparatus: (**a**) batch SFEE (b-SFEE), (**b**) continuous SFEE (c-SFEE), and (**c**) c-SFEE with packed column (c-SFEE-PC). Reprinted from [1] (Elsevier 2009) and [51] (Elsevier 2011) with permission.

#### 2.2.2. Mechanism of SFEE

##### Drawbacks of Conventional Micro-/Nanoparticle Solidification Processes

In conventional micro-/nanoparticle solidification processes, after preparing the emulsion, heating at high temperatures, or using of a large amount of extraction solvent, an additional separation process may be required to remove the used organic solvent for solidification into the desired micro-/nanoparticles. Solidification through solvent removal can significantly affect several critical physical properties, such as pore size and volume, particle density, surface energy, and particle size and distribution. These physical properties ultimately determine pharmaceutical performance properties such as encapsulation efficiency (EE), drug loading capacity, drug release profile (burst release), and in vivo performance [11]. In general, rapid fabrication of particles through improved extraction efficiency is known to be a very useful approach to improve these properties and develop an efficient manufacturing technology for targeted particle size with narrow PSD and low agglomeration within the nano to micro range. This important aspect can be explained well by taking a representative example of *W/O/W* solvent removal, which is widely used in the development of long-acting sustained-release microspheres for biopharmaceuticals (Figure 5). It is well known that one of the most critical factors for the preparation of desired microparticles is solvent removal. In the solvent evaporation/extraction process, *W/O* emulsion droplets come in contact with a large volume of the continuous aqueous phase; subsequently, fabrication into solid particles is initiated as the organic solvent diffuses into the continuous phase. However, using only this simple diffusion mechanism for solvent removal can increase the possibility of fatal problems such as low EE and agglomeration between particles because the solvent removal rate is too slow. This slow solvent removal rate can also result in the slow hardening of the capsule shell or matrix particles, leading to the diffusion of protein molecules into the continuous phase. This phenomenon may cause the problem of initial burst release due to the exposure of drug molecules to the surface of solid particles [11,23,26,28,30,52]. Therefore, to overcome abovementioned limitations and the success of the emulsion method for micro-/nanoparticle drug delivery systems, it is necessary to select the optimal solidification process considering the efficiency of solvent removal.

##### Excellent Solvent Removal and Solidification Mechanism of SFEE

In several previous reports, the detailed processing mechanism and extraction behavior of SFEE have been well defined and described. This is briefly summarized and explained below. 

The mechanism of micro-/nanoparticle formation from emulsions during the SFEE process is shown in Figure 6 as proposed by Della Porta et al. [44]. When the starting emulsion containing drug with or without excipients (e.g., polymers and/or surfactant) is introduced into the high-pressure vessel filled with the SCF, the SCF in contact with the emulsion extracts the organic solvent from the emulsion via mass transfer using two main routes that have been named by Della Porta et al.: ‘‘direct solvent extraction’’ and ‘‘indirect solvent extraction” (Figure 6). The direct solvent extraction route includes the mass transfer of the organic solvent by SCF directly diffused into the internal oil phase of the emulsion droplet. In contrast, indirect solvent extraction refers to the subsequent mass transfer of the organic solvent equilibrated in the outer aqueous phase into SCF, following the diffusion of the organic solvent into the continuous phase. 

For a more detailed explanation, it is necessary to understand the characteristic changes in the volume and appearance of organic solvent droplets as an oil phase suspended in aqueous phases overtime during the extraction process at the supercritical state [53]. Mattea et al. reported that after pressurization by SC-CO_2_, the volume of dichloromethane (DCM) droplet suspended in water increased (swelling of the droplet) due to the direct diffusion of SC-CO_2_. This observation indicates direct solvent extraction (Figure 6a,c). During this direct solvent extraction, the dissolved solute in the DCM droplet is precipitated by SC-CO_2_ as an antisolvent. After the droplet volume has reached its maximum limit, the volume of the droplet decreases rapidly (shrinking of the droplet). This shrinking may result in particles smaller than the starting emulsion droplets. It has been suggested that this is probably because DCM diffuses out into the external aqueous phase and is rapidly and continuously removed by SC-CO_2_. This solvent extraction mechanism involves indirect solvent extraction (Figure 6b,c). 

The extraction mechanisms of these two pathways are not independent of each other and usually coexist during the SFEE process. The synergistic effect of the combination of both extraction pathways can induce excellent mass transfer efficiency of SCF, resulting in the fast transformation of solute into solid due to the higher supersaturation inside the droplet (without a chance to be agglomerated by the contact between droplets). This favorable effect can lead to the reproducibility of a non-aggregated spherical particle shape with more uniform particles in a narrow PSD.

## 3. SFEE Application Cases

There are many applications of SFEE technology for various purposes, such as the development of improved drug delivery systems (e.g., microencapsulation for controlled release, nanoparticles for improved drug delivery, pulmonary drug delivery, polymeric gene delivery, tissue engineering, and nanoparticles of inorganic materials), solubilization via nanoparticles of poorly water-soluble drugs, physicochemical stabilization, and solidification of liquid material (Table 2).

### 3.1. Drug Delivery System 

#### 3.1.1. Microencapsulation for Controlled Release

As mentioned above, *W/O/W* and *S/O/W* methods for emulsion preparation are well suited to encapsulate hydrophilic and water-soluble drugs, including both chemical and biomolecules. In contrast, the *O/W* method is ideal for hydrophobic and poorly water-soluble drugs [27]. First, we would like to introduce research cases of microparticles for controlled release prepared using SFEE after preparing an *O/W* emulsion of poorly water-soluble drugs.

Chattopadhyay et al. of Ferro Corporation aimed to prepare sustained-release microsphere formulations using SFEE [54]. Two hydrophobic drugs, indomethacin and ketoprofen were selected as model drugs. Poly (lactic/glycolic) acid (PLGA) and Eudragit RS were used as the polymers for encapsulation. A high-speed dispersator and high-pressure homogenizer were used for the emulsion droplet sizes in the micro and nanometer ranges, respectively. The particle sizes of the micro and nanoparticles obtained in stable aqueous suspensions ranged from 0.1 to 2 μm. The solvent residue (ethyl acetate, EA) in the suspension obtained was below 50 ppm. The intrinsic dissolution kinetic coefficients of drugs from drug-polymer microparticle prepared using SFEE were 2–4 times lower than those of raw material particles. It was also found that the diameter of the emulsion droplet, the concentration of the polymer and the drug in the solution, and the ratio of the solvent in the emulsion determine the size of the microsphere.

Della Porta et al., colleagues of Professor Reverchon’s research team in Italy, reported that they prepared a sustained-release PLGA/piroxicam microsphere system using SC-CO2 extraction from *O/W* emulsions [24]. The effect of the PLGA concentration in the *O/W* emulsion, extraction pressure and temperature in the supercritical state, flow rate of SC-CO_2_, and extraction time on the PSD and residual solvent level were investigated. The PLGA microspheres obtained were spherical with narrow size distributions ranging between 1 and 3 μm in mean particle sizes. It was also shown that the SFEE process can prevent coalescence or aggregation of emulsion droplets and particles, which generally occur in conventional SE. In addition, a short extraction time of approximately 30 min and residual EA level lower than 40 ppm are used. In contrast, the content of the EA residue in the microparticles prepared using the SE process was 500 ppm. This indicates that the SFEE process is much faster and more efficient than the conventional SE process. Furthermore, the drug EE ranged from 90% to 95%, and no burst drug release and a sustained release were observed. 

This research team conducted further studies to prepare microspheres using the SFE of *O/W* and *W/O/W* emulsions for piroxicam and diclofenac, respectively [43]. The microspheres obtained had mean particle sizes in the range of 1–3 μm. It was confirmed again that the microspheres prepared via SFEE were produced with better reproduction of the original emulsion droplets. In addition, the measured concentration of the residual solvent was 10 ppm, whereas it was 600 ppm for the conventional SE process. They suggested that the extraction rate of SFEE is a faster process with more efficiency, which prevents particle aggregation and better control of particle formation within a narrow range of PSD, in contrast to the aggregation found in microspheres prepared using SE. Moreover, improved EE and sustained release profiles were observed for the microspheres prepared using SFEE. In particular, the microspheres obtained using SFEE showed significantly higher EEs, 88% for microspheres from *W/O/W* emulsion, and 97% for microspheres from *O/W* emulsion, when compared with the results of SE, which showed EEs of 30% and 70%, respectively.

Della Porta at al. developed a c-SFEE process using a countercurrent packed column tower (c-SFEE-PC) as a patented new technology for microparticle production (Figure 4c) and then reported the research results regarding their c-SFEE technology and its applications [44]. The proposed c-SFEE process was applied to prepare PLGA microparticles without drug material. The influences of c-SFEE process parameters, including temperature, pressure, flow rate, and its ratio on the recovery of the processed material and size distribution of the microparticle obtained, were investigated and compared with those of the b-SFEE and SE processes. It was shown that the c-SFEE-processed PLGA microparticles had a more uniform PSD in the range of 1–3 μm than that of b-SFEE, even though the internal structure of the polymer microparticles was almost identical to that prepared using the conventional SE process. They stated that the narrow PSDs obtained through more fine control of particle formation is a major advantage of this new c-SFEE technology, and the extraction time could be completed in just a few minutes to comply with the residual solvent regulation. It was suggested that this improved result is due to the significantly improved extraction efficiency via the large contact area between the emulsion and SC-CO_2_ in the packed column. 

This c-SFEE-PC process was applied by Falco et al. to develop a local injection formulation (PLGA microsphere formulation of hydrocortisone acetate) for prolonged action [55]. Two microsphere formulations were prepared from *W/O/W* and *S/O/W* emulsions using c-SFEE. The microspheres obtained showed mean particle sizes in the range of 1–5 μm with excellent EE between 75% and 80%. They stated that high EE for hydrophobic drugs, especially corticosteroids, can be achieved by rapid drug crystallization from the internal phase of the emulsion in combination with fast and continuous emulsion processing. Interestingly, the drug release profile was dependent on the types of emulsion used. Prolonged drug release up to approximately 15 days and a relatively faster release up to about 6 days were observed for microspheres fabricated from *S/O/W* and *W/O/W* emulsions, respectively. Based on these results, it was suggested that c-SFEE equipped with a high-pressure packed column is a promising pharmaceutical particle formation process with higher processing efficiency and product uniformity even with a small plant. Furthermore, it was suggested that it could be adopted as an option for good manufacturing practices (GMP) production in sterilized pre-filled syringes.

Unlike the hydrophobic drugs mentioned above, the following contents are research cases of SFEE application to prepare microparticles containing hydrophilic and water-soluble chemical drugs and biomolecules. 

Kluge et al. used the b-SFEE process to produce micro-/nanoparticles consisting of a PLGA biodegradable polymer and lysozyme as a sensitive protein biomaterial [25]. The formulation and process parameters, such as the emulsion type, polymer concentration, and mechanical force for stirring influence the preparation of emulsions. The control of these parameters resulted in reproducible spherical solid particles with a mean particle size in the range of 100 nm to a few μm with a very narrow PSD. In particular, it was found that the choice of emulsion type is a very important factor for improving EE. Three different emulsion types, *W/O/W*, *S/O/W* and in situ *S/O/W*, were evaluated to study their effect on EE, and the EE results obtained were 11%, 37% and 49%, respectively. In addition, much smaller particles were obtained using the in situ *S/O/W* emulsion method. They concluded that the SFEE has great potential as a scalable commercial process for the manufacturing of pharmaceutical micro-/nanoparticles, with many advantages such as low solvent residue and high purity, moderate operating temperatures, and reasonable consumption of environmentally and economically friendly CO_2_.

Professor Reverchon’s research team has also led the research and development of commercially available microparticle manufacturing technology using c-SFEE of various water-soluble biomolecule drugs.

Falco et al. reported that c-SFEE-PC was applied for PLGA microencapsulation of insulin, as a model drug for protein, a representative hydrophilic biomolecule. The starting emulsion was *W/O/W* [56,57]. The fundamental fluid dynamic mechanism of a packed column tower in a c-SFEE apparatus was investigated to define and calculate the loading condition for efficient mass transfer and flooding conditions, and its correlation with the density difference between the two different phases from the experimental data. Here, the flooding condition was defined as a point where the column is no longer operative so that the liquid cannot flow down through the column owing excessive increases in gas or liquid velocity overloading conditions. In addition, they evaluated the size, morphology, and insulin EE of the microspheres obtained. Non-collapsed spherical microparticles were generated with an acceptable residual solvent level (EA) lower than approximately 600 ppm. The mean particle size ranged from 1.8 to 4.8 μm, and EE was about 70%. In the drug release test of the prepared PLGA microspherse in PBS medium, a burst insulin release appeared on the first day, followed by an extended release for 24 days. For a microsphere formulation with a higher loading of insulin, faster burst release, almost 90% of loaded insulin, was observed on the first day. Importantly, it was also revealed that the SC condition at relatively high pressure did not lead to any denaturation and/or degradation of thermolabile molecules, such as proteins.

Campardelli et al. showed that c-SFEE can be applied to the preparation of biodegradable polymer (PLGA or poly-lactic acid, PLA) particles containing various polypeptides and proteins. Bovine serum albumin (BSA) and human insulin-like growth factor (h-IGF) were used as model proteins and polypeptides [58]. These water-soluble biomolecules were dissolved in the internal water phase and then used to prepare *W/O/W* emulsions for injection into the c-SFEE process. It has been shown that both nanoparticles and microparticles can be successfully fabricated as designed from nanoscale in the range of 200–400 nm to microscale in the range of 1–4 μm with narrow PSD owing to the fast extraction rate, hence minimizing aggregation of droplets. This was confirmed by the results that the PSDs of the solid particles obtained were very similar to those of the emulsion droplets. In addition, EA used as an organic solvent in the oil phase was eliminated very quickly at 37 °C and 100 bar, and it was suggested that this mild supercritical condition can prevent the degradation of proteins and peptides.

They also applied c-SFEE to the development of injectable polymer microparticles that encapsulated water-soluble biomolecules and chemical drugs simultaneously for controlled release [59]. Several combinations of chitosan, PLGA, and hydroxyapatite were used for the polymer composition that constitutes a matrix of particles or an outer layer of microcapsules. Teriparatide and gentamicin sulfate were selected as model water-soluble biomolecules and chemical drugs, and this combination therapy was designed to heal complex fracture or treat local bone pathologies in elderly patients. It was shown that the mean particle size of the c-SFEE processed microcapsules was in the range 1.4–2.2 μm, and EE was up to 90%. The prepared microcapsules showed well-controlled drug release for 15–20 days in PBS at 37 °C. Interestingly, it was observed that hydroxyapatite-added microcapsules delayed the release of teriparatide. In addition, the initial burst release of the drug was improved significantly through the addition of chitosan to the polymer composition.

#### 3.1.2. Nanoparticle for Improved Drug Delivery 

It is well known that nanoparticle systems of drug can improve the bioavailability and drug delivery of the desired target by increasing absorption and/or prolonged residence in the body, thereby enhancing drug efficacy, increasing tolerability, and reducing the risks of toxicity [74,75,76,77,78,79]. In addition, nanoparticles encapsulated with polymers can be used to prevent the degradation of drugs. Nanoparticles with biodegradable polymer are generally used to improve the therapeutic efficacy of various medicinal drugs, including chemicals and biomolecules ranging from water-soluble to insoluble substances [10,80,81,82].

These typical advantageous characteristics of drug nanoparticles can be modified and optimized by controlling the particle size and surface properties such as porosity, charge, and hydrophilicity or hydrophobicity. Among these various factors, the most important properties are particle size and its uniformity because the penetration of nanoparticles across the biological membrane can be dependent on the particle size [10,83].

To prepare polymer nanoparticles with desirable properties, especially narrow PSD and high EE with low loss of drug, it is very important to select a suitable formulation and manufacturing process. Starting from an emulsion, solidification via removal of an organic solvent has been widely used as a method for preparing solid nanoparticles. In this method, the diameters and distributions of the prepared nanoparticles were generally very similar to those of the corresponding emulsion droplets. If the nano-sized droplets can be formed through the control of the various process and formulation parameters in the emulsion step, then the organic solvent is quickly removed, and the polymer can be fabricated as it is in the nano-size of the droplets without expansion of the droplets. The SFEE can be applied to maximize the efficiency of this nanoparticle manufacturing mechanism, and the application cases are described below [10].

Della Porta et al. (Professor Reverchon’s research team) reported that monodisperse polymer nanoparticles were prepared using the c-SFEE-PC process [60]. PLA, PLGA, and polycaprolactone (PCL) were used as the biodegradable polymers. For the *O/W* emulsion, acetone was used as the oil phase, and glycerol was included in the outer aqueous phase. They showed that monodisperse nanoparticles in the range of 200–400 nm with a polydispersity index lower than 0.1 nm were prepared through the c-SFEE process at 38 °C and 80 bar. The measured residual solvent levels for all particles obtained were as low as 500 ppm. It was shown that the reduced emulsion processing time and particle aggregation by the c-SFEE process resulted in a well-controlled PSD. In addition, they stated that direct optimization of the emulsion droplet size distribution makes it possible to obtain monodisperse nanoparticles together with efficient c-SEEE process conditions. 

Mitchell et al. showed that b-SFEE can be successfully used for the preparation of monodisperse PCL nanoparticles without particle agglomeration in a short process time (only 1 h) when compared with the conventional solvent extraction method at atmospheric pressure [61]. The starting emulsion type was *O/W*, and nanoemulsions were prepared using a spontaneous emulsification mechanism. The external water phase consisted of deionized water, while the oil phase consisted of PCL and Tween 80 in acetone. The particle size was reduced by decreasing the concentration of PCL or increasing the surfactant concentration in the emulsion. Spherical nanoparticles with particle sizes between 200 and 500 nm were obtained in the formulation range of 2:1–16:1 of PCL-Tween 80 weight ratio (*w/w*). They stated that the polymer concentration was the most important factor in determining the final nanoparticle size. 

Murakami et al. suggested the application of microfluidic slug-flow to b-SFEE for reduced cost and time in nanoparticle manufacturing [62]. The starting emulsion type was *O*(*EA)/W*, and polyvinyl alcohol (PVA) was used as the model polymer. PVA nanoparticles in suspensions were obtained in the particle size range of 10–20 nm without organic solvent residue. The influence of the space time and the contact area between the emulsion and SC-CO_2_ on the extraction efficiencies during SFEE was investigated. Interestingly, it was found that the hydrophobic PVA in the emulsion led to enhanced extraction efficiencies compared with the hydrophilic PVA. In addition, they stated that mass transfer from the oil droplet to the aqueous phase is the rate-determining step of the SFEE [15]. They applied the SFEE in microfluidic slug flow to prepare nanoparticles of ibuprofen-PVA as. In addition, the nanoparticles obtained were functionalized with chitosan through a combination with the chitosan dissolution technique using SC-CO_2_. A phase behavior study was conducted to observe whether chitosan could be dissolved in the aqueous phase under SC-CO_2_ and to understand the mass transfer mechanism under extraction at SC conditions. It was confirmed that chitosan can be dissolved in the water phase after 30 min exposure in SC conditions due to decreased pH by SC-CO_2_, and this phenomenon can be successfully used for nanoparticle surface functionalization with chitosan. The prepared nanoparticles of ibuprofen-PVA functionalized with chitosan showed an average diameter of approximately 200–300 nm and a positively charged surface, hence improving the bio-adhesive property.

Giufrida et al. applied b-SFEE to prepare hydrophobic medroxyprogesterone acetate-encapsulated poly(3-hydroxybutyrate-co-3-hydroxyvalerate) nanoparticles [63]. *O*(DCM)/*W*(PVA) prepared using ultrasonication was used as the starting emulsion. The mean particle dimeter of the nanoparticles obtained using various molecular weights of the polymer were between 183 nm and 850 nm. The sizes of the nanoparticles prepared via SFEE were smaller than those obtained via SE of the emulsion. In addition, a high drug EE of approximately 70% and low toxicity profiles were observed for nanoparticles prepared via SFEE. Furthermore, it was shown that 35% of the drug was released within 18 h with first-order release kinetic. 

#### 3.1.3. Pulmonary Drug Delivery

Efficient pulmonary drug delivery can typically be achieved through inhalation of particles below 5 μm for the deep deposition of drugs to the central and peripheral regions of the lung. Thus, in the development of medical products for lung inhalation, it is important to uniformly produce micro-sized particles within a narrow PSD suitable for lung delivery [34,84,85,86].

Chattopadhyay et al. reported that c-SFEE can be applied to prepare solid lipid nanoparticles (SLNs) in nanosuspension systems for pulmonary delivery [64]. They used indomethacin and ketoprofen as model drugs with poor water solubility, and these drugs were formulated with lipid excipients such as tripalmitin, tristearin, and Gelucire 50/13, at a drug loading ratio of 10–20% *w/w*. Physical characterization showed that the nanosuspension was stable, although the drugs in the SLN were in an amorphous state. The measured solvent residue was less than 20 ppm. The mean particle size of the SLNs obtained and measured using the laser diffraction method was below 50 nm with a narrow PSD, which showed that significantly reduced particle size was achieved via c-SFEE with consistent production when compared with other nanoparticle formulations obtained by previously reported technologies. The prepared SLN suspension was sealed in AERx strips and used to investigate the aerosol PSDs and feasibility of lung delivery via aerosolization of the c-SFEE-processed SLN using the AERX™ aerosol delivery system (Aradigm Corporation, Hayward, CA, USA) combined with an Andersen cascade impactor. The fraction of particles less than 3.5 μm aerodynamic particle size was greater than 90% of the total aerosolized particles. The consistent aerosol PSD results indicated that the SLN suspension prepared using c-SFEE can be successfully aerosolized with minimized particle deposition in the oropharyngeal region and enhanced deep lung deposition. In addition, it was also shown that the type of starting emulsion and its droplet size were the major parameters for controlling the particle diameter.

#### 3.1.4. Polymeric Gene Delivery

Recently, the therapeutic use of gene medicine such as plasmid DNA (pDNA) has been well studied, but there is a limit to the commercial development of pharmaceutical delivery systems with both safety and efficacy. Biodegradable polymer matrix nanoparticles have been commonly used as nonviral gene delivery vectors in nanoparticle systems, but there are still several limitations, such as low loading efficiency and solvent residue, that cause fatal damage to medicinal genes. To overcome these hurdles, Mayo et al. studied the applicability of b-SFEE for the preparation of PLGA-nanoparticle formulations containing pDNA [65]. pFlt23K or pEGFP were used as model plasmid DNAs, and *W/O(EA)/W* as a starting emulsion was prepared using sonication. The nanoparticles obtained with a size range of approximately 150–350 nm had a spherical shape, as observed using transmission electron microscopy (TEM) image analysis. In addition, c-SFEE resulted in a high pDNA loading efficiency of above 98% and a low organic solvent residue below 50 ppm. They suggested that these results were due to the fast solidification and excellent solvent extraction efficiency of the c-SFEE. Moreover, it was shown that the incorporated plasmid was effectively released from the polymer nanoparticles. The effective in vitro transfection and significantly reduced VEGF secretion without cytotoxicity for the obtained pFlt23K nanoparticles indicated that this nanoparticle formulation has potential for the treatment of neovascular disorders.

#### 3.1.5. Tissue Engineering

Nano-microencapsulation of biodegradable regenerative medicine has been used in tissue engineering. The abovementioned conventional encapsulation techniques have several problems, such as low EE and reduced cell viability [67]. For this tissue engineering, several application cases of SFEE for encapsulating chemical materials or biomedicines such as cells will be introduced below.

Palazzo et al. used c-SFEE-PC processing of *W/O/W* emulsions for the PLGA microencapsulation of two human growth factors, which are small biomolecules widely used in tissue engineering for the commitment of stem cells via inducing cell growth, proliferation and differentiation [66]. The variation in molecular weight of the polymer, composition of the surfactant, and mixing speed resulted in particle sizes ranging from 0.4–3 μm and drug loading capacity sizes ranging from 3–7 μg/g. The release of growth factors from microparticles was sustained for more than 25 days. In addition, the reduced cytotoxicity through low reactivity on human peripheral blood mononuclear indicated its potential as a safe biomedical device. These results demonstrated the applicability of SFEE technology for the development of 3D assembled matrices for tissue engineering such as bioengineered scaffolds.

Recently, it has been suggested that genetically engineered bacteria capable of producing various substances with therapeutic effects can be applied as delivery vectors for various antigens and biomaterials. In this respect, the availability of SFEE as a useful technology for the manufacture of biodegradable microdevices incorporating prokaryotic cells for tissue engineering was suggested by Della Porta et al. [67]. They investigated the application of SFEE for the microencapsulation of *Lactobacillus acidophilus* as a model bacterium in PLGA microparticles. The starting emulsion type was a *W/OW* double emulsion, and EA was used as an organic solvent in the oil phase. PLGA microparticles with a mean diameter of 20 μm were successfully prepared via SFEE process for 30 min at 90 bar and 37 °C. The bacterial loading was 0.6% (*w/w*), and EE was approximately 80%. Scanning electron microscope/energy-dispersive X-ray spectroscopy (SEM-EDX) and particle size analyses showed the regular size and morphology of microparticles were fabricated with well-entrapped cells inside the particles rather than the outer surface. In addition, the availability of sustained release properties was shown for longer efficacy in the implanted area. It was concluded that the SFEE technology with low cell viability can be mainly used to develop the killed vectors that can act as biological signals but reduce side effects.

In addition to these medical therapeutic purposes, it was reported that the SFEE can be used to produce nano-/microparticles encapsulating structural health-monitoring agents to detect the structural defects in the damaged region and promote the release of self-healing material [68]. 

#### 3.1.6. Nanoparticles of Inorganic Materials

Inorganic nanoparticles have been applied widely in various fields such as pigments, coatings, packaging, photocatalytic applications, diagnostic applications, and biological and medical applications. However, high density and unique surface characteristics, which are specific properties of nanoparticles, are highly likely to cause serious problems in the physical stability of colloidal systems such as sedimentation and agglomeration. Several pharmaceutical technologies such as co-precipitation or encapsulation with polymers have been proposed as a solution to improve the colloidal stability of inorganic nanoparticles [87]. Campardelli et al. applied c-SFEE-PC in the preparation of TiO_2_ nanoparticles [69]. They first prepared a stabilized TiO_2_ colloidal nanoparticle suspension in an ethanoic aqueous solution using a microwave-assisted precipitation method from titanium propoxide solutions, and the particle size obtained was approximately 15 nm. Then, the nanoparticles were used to prepare two types of starting emulsions, *W/O/W* and *S/O/W*. The oil phase consisted of EA and PLA as the organic solvent and polymer, respectively. Through the c-SFEE process of the double emulsions, TiO_2_-PLA nanoparticles with a mean particle size of 200–900 nm were obtained with a high EE of up to approximately 90%. TEM and X-ray photoelectron spectroscopy analyses confirmed that the TiO_2_ nanoparticles were uniformly dispersed in the PLA matrix. Moreover, it was shown that the prepared TiO_2_-PLA nanoparticles maintained their photocatalytic bactericidal activity against *Staphylococcus aureus* without reduction due to the formation of the outer polymer shell barrier. They concluded that the production of inorganic polymer nanocomposites using c-SFEE can allow the manufacture of biocompatible medicinal devices.

In addition, the c-SFEE process was successfully applied to produce biocompatible magnetite nanoparticles as a magnetic resonance imaging contrast agent for diagnostic applications [70]. Fe_3_O_4_ nanocrystals were prepared using a co-precipitation method via stabilization with ricinoleic acid. For the preparation of the *S/O/W* starting emulsion, the magnetic nanocrystals obtained were homogeneously dispersed in the oil phase of the polymer (PLGA)-DCM solution and then emulsified with an outer water phase through sonication under magnetic stirring. The c-SFEE led the formation of a stable nano-suspension consisting of the Fe_3_O_4_-PLGA nanocomposite via efficient DCM removal. The mean particle size was 140–230 nm by changing the emulsifier PVA concentration from 1–4%. Interestingly, it was observed that the morphology of the nanoparticles showed the characteristics of Janus particles, which are special types of nanoparticles with surfaces having two or more different physical characteristics. This indicated that Fe_3_O_4_ accumulated at the one hemisphere surface of the nanoparticle as a large cluster. These results show that c-SFEE is a promising technology for commercial-scale production of high-quality magnetite nanoparticles via better control of morphology and size. 

### 3.2. Solubilization via Nanoparticles of Poorly Water-Soluble Drugs

Particle size reduction techniques are commonly used to enhance the dissolution rate and oral drug absorption of poorly water-soluble drugs included in biopharmaceutical classification system (BCS) class II. Generally, reducing the particle size at the nanoscale can increase the surface area, thereby leading to an improved dissolution rate and solubility and, consequently, a faster onset of action and higher oral bioavailability [88,89,90,91,92].

Shekunov et al. evaluated c-SFEE technology for the preparation of nanoparticles of poorly water-soluble drugs such as cholesterol acetate, griseofulvin, and megestrol acetate [71]. The *O/W* type starting emulsion was prepared by dissolving the drugs in organic solvents, such as EA, toluene, or DCM, and then emulsified in the outer water phase, including some stabilizers such as surfactants or polymers using high-pressure homogenization. In contrast to long acicular microparticles with mean diameters of 20–200 μm obtained via the SAS process, regular and uniform prismatic crystals of 100–1000 nm with crystallinity in nature were obtained via the c-SFEE process. The measured solvent residue was pharmaceutically acceptable at the ppm level (<60 ppm). In addition, the CPPs for nanoparticle precipitation using c-SFEE are drug concentration, amount of organic solvent, and droplet size of the emulsion. The prepared nanoparticles showed a significantly improved dissolution rate (approximately 5–10 times) when compared with micron-sized particles. It was shown the c-SFEE process can offer an important advantage for particle size reduction with more flexibility in the fine control of physical properties such as crystallinity, particle morphology, and various surface properties than those of the SAS process. Consequently, it was suggested that the c-SFEE process is a promising high-purity nanoparticle manufacturing technology with a short production time and ease of scale-up, although it may be difficult to maintain the emulsion stability during extraction.

### 3.3. Physicochemical Stabilization

In the manufacture of drug or drug-polymer micro/-nanoparticles using various pharmaceutical technologies, it is frequently observed that the prepared micro/-nanoparticles have a reduced crystallinity or amorphous state that may be physically and chemically unstable. For the physicochemical stabilization of micro/-nanoparticles, Kluge et al. used the b-SFEE process as a nanoencapsulation technology [72]. The model drug and polymer were ketoprofen and PLGA, respectively. To prepare the *O/W* emulsion for injection into the SFEE process, they were dissolved in EA and then emulsified in the outer aqueous phase using ultrasonication. The drug-polymer nanoparticles (100–200 nm) showed a mono-phase at a certain equilibrium level where the drug was dissolved in the polymer. Moreover, drug nucleation was not observed during 1 week of storage because the supersaturation level of drug-polymer nanoparticles was too low. These results indicate that an overloaded metastable drug-polymer nanoparticle formulation can be achieved [72]. In addition, applications of SFEE for the production of micro-nanoparticles with improved chemical stability in various antioxidants have been reported.

### 3.4. Solidification of Liquid Drug

Fat-soluble drugs in the liquid phase are often prepared as micro-/together with solid polymers to improve ease of administration, physicochemical stability, and drug release control. In this respect, Prieto and Calvo used the b-SFEE process to encapsulate hydrophobic vitamin E with a biocompatible polymer, PCL [73]. Non-aggregated spherical nanoparticles with a core-shell structure were obtained in the particle size range of 8–276 nm with a narrow PSD. In addition, a high EE of approximately 90% and low organic solvent residue of approximately 50 ppm were observed for the prepared vitamin E-PCL nanoparticles. It was shown that the SFEE process available at relatively low temperatures is suitable for the nanoparticle solidification of heat-sensitive liquid or low-melting drug materials. 

## 4. Expert Opinions and Perspectives

Although many conventional micro/nanoparticle manufacturing processes described above are currently used in the production of final drug products in the pharmaceutical industry, they still have many problems. Particularly, the manufacturing process and properties of the particles obtained are extensively affected by the microsphere injection dosage due to a sustained drug release. The limitations of the particle production process are complex, multistep, and affect the entire process, including discontinuous/discontinued step, large amounts of toxic solvents, drug decomposition potential due to high-temperature conditions for solvent removal, long production time and inefficiency, consequent increase in manufacturing cost, and low productivity. In terms of pharmaceutical formulations and biopharmaceutics, nonuniform particle size distribution, low EE, and increased risk of side effects due to excessive initial burst release are unresolved problems. 

The SFEE process proposed to solve these problems can improve the pharmaceutical properties of medicinal micro/nanoparticles by exploiting the inherent advantages of SCF to achieve superior solvent removal efficiency and fast solidification. Therefore, it is expected that SFEE technology will be an excellent technology for manufacturing solid micro/nanoparticles with superior quality to those of existing particle manufacturing technologies. In particular, due to the mild process temperature, SFEE can be the first choice for particle manufacturing of biopharmaceuticals when considering stability. 

However, this promising SFEE technology had several problems. In particular, there were two major limitations in developing SFEE as a commercial particle production process. First, the initial batch-type equipment was considered difficult to scale up and industrialize because of the requirements of large plant areas and facilities and the burden of initial equipment investment for these new technologies. This problem was overcome by the development of the c-SFEE-PC. Unlike the previously reported b-SFEE process, c-SFEE-PC technology and its equipment have already been developed and do not require large plant facilities for continuous mass production of pharmaceutical micro/nanoparticles. The second problem was the difficulty in developing a commercially available process because emulsion formation and solidification via solvent removal were typically two separate processes. This problem has also been addressed by the development of continuous emulsion production equipment, which can be used together with c-SFEE equipment for a continuous manufacturing process. In particular, because emulsification and drying equipment already installed in pharmaceutical factories can be used in the SFEE process, it also has the advantage of reducing the time and cost of designing and manufacturing production facilities. In addition, several studies have reported the reproducibility of uniform particle production using c-SFEE. Therefore, the potential for the successful commercialization of SFEE technology has already been demonstrated through the introduction of improved c-SFEE processes developed by several leading researchers. 

Although there have been reports that the improved SFEE process has many advantages for various pharmaceutical applications, there is a need to develop it into a GMP-compliant facility with product reproducibility to achieve commercial mass production. In the pharmaceutical manufacturing processes, GMP is necessary to regulate the potential chemical and mechanical hazards [40]. However, large-scale research to establish plant facilities of SFEE suitable for GMP that can produce micro-/nanoparticles on an actual commercial scale is insufficient and has not yet been completed. There have been few reports on the effects of various process variables on the output. However, even quality by design (QbD)-based approach studies at the laboratory scale are not enough, as is the case with the pilot or commercial scale. The lack of the fundamental understanding of the theory of particle formation in SFEE following the lack of scientific analysis of experimental data for controlling fine particle formation from lab scale to scale-up can be compensated by studies using QbD-based approaches [93,94,95,96,97,98]. Thus, to achieve successful GMP scale-up and commercial mass production technology, it is essential to establish validated and robust technology through a scientific and statistical strategy based on the QbD approach [99,100]. In contrast to SFEE process in pharmaceutical field, extraction technology using SCF has already been used commercially in the food industry to extract effective active ingredients from natural raw materials or to selectively remove unwanted substances [101,102,103,104,105]. In the food industry field, there have been many QbD-based research studies for the scale-up and validation of the supercritical extraction process, and finally, successful commercialization has been achieved based on fundamental understanding [106,107,108,109,110,111]. Furthermore, cost and economic analysis studies of the pilot and industrial scale operations have been also performed [112,113]. Through these multilateral studies in the food science field, several supercritical extraction applications mainly for food products, ingredients, and nutraceuticals (e.g., extraction of caffeine from coffee and tea, purification of nutraceutical materials such as aroma oil, essential fatty acids, natural vitamins, and removal of pesticides from agricultural product) performed in lab-scale have eventually scaled up to production on pilot scale and even commercial scale [114]. Although the solvent-removal and particle-formation mechanisms are completely different from those of SFEE, PGSS, and pressurized gas expanded (PGX) technologies, SCF-based particle formation processes have also been successfully commercialized in the food industry [113,114,115,116,117,118,119]. Fortunately, we can take advantage of these successful commercialization examples in industrial food field and use them for commercial development in the pharmaceutical field. It is expected that the large number of SC-CO_2_ extraction plants and facilities can be redesigned and/or expanded to suit SFEE equipment [114]. Even if the more stringent GMP regulations for pharmaceuticals are required than regulations for food products, its successful application in pharmaceutical production will be a matter of time. In addition, the amount of SC-CO_2_ used and its recycling should also be considered in scale-up of the SFEE process. It was reported that the ratio of SCF to solute in a range of 1 to 10,000 (*w/w*) for various SCF-based processes at lab-scale can be reduced to 50 (*w/w*) at commercial scale by recycling of used SCF [120]. In addition, a design of a step for separation of the used organic solvent and the SCF also should be considered for recycling of SCF during SFEE process.

Another problem is that for all emulsion-based particle manufacturing technologies, including SFEE, the final products are prepared in the form of a suspension; therefore, an additional drying process is required to obtain the final product in the dried solid form. This additional drying process can lead to agglomeration of particles, an increase in particle size, and physicochemical destabilization of drugs. In addition, because this is a separate step from the solidification process, it may be difficult to develop it as a continuous process. Consequently, several novel attempts have been made to combine SD or spray-freeze drying methods after suspension generation, but there is still no ideal technology and research for an improved process is ongoing. Therefore, it would be ideal to develop a GMP plant comprising particle drying equipment, which operates in line with the process wherein the suspension is obtained through c-SFEE. A combination of the SFEE processes with several techniques using SCF as an atomization agent (e.g., SAA and SASD) could be a good approach to achieve this purpose. Another important thing to consider in order to complete c-SFEE as a dried particle formation technique is the particle collection method and apparatus for safe handling of micro-/nanoparticles in the lab, pilot, or industrial scale to meet the GMP guidelines [114]. Several fine-particle collection methods can be applied using inertial effect (e.g., cyclones and impact chambers), electrostatic effect, or washing and filtration [120]. 

There has been no clinical study case of pharmaceutical product developed using SFEE yet. This will be achieved after the establishment of the SFEE process suitable for GMP. Considering the current limitations of SFEE mentioned above, it may be a long road to achieve the ultimate goal of deriving successful development cases from the clinical stage to the final drug product. Nevertheless, many studies in SFEE technology have focused on and will focus on how to overcome the abovementioned current limitations of SFEE. Moreover, compared with conventional micro/nanoparticle manufacturing processes that use a large number of organic solvents, the SFEE process is an environmentally benign green technology because it uses a green solvent (SC-CO_2_), minimizes the use of toxic organic solvents, and has a positive ecological impact, thereby reducing the potentially toxic and hazardous materials and by-products. These eco-friendly features correspond to the characteristics required as future technologies from an ecological point of view. Consequently, it is expected that such continued research of SFEE technology will make it a promising particle formation technology that can be commercialized and successfully applied to actual micro-/nano medicines in the near future. In addition, the first successful commercialization case of SFEE can promote it to be widely used in the pharmaceutical field as well as various applications in the chemical fields, including food industry.

## 5. Concluding Remarks

It was confirmed through various pharmaceutical applications of SFEE (described in detail above) that SFEE can produce uniform micro/nanoparticles for both hydrophilic and hydrophobic drug substances. The wide applicability of the promising SFEE technology for the production of pharmaceutical micro-/nanoparticles and their physicochemical and biopharmaceutical advantages are thoroughly described in this review. In summary, the excellent extraction efficiency of SCF via fast mass transfer and high solvent power can avoid the main drawbacks of several traditional particle formation technologies such as SE, solvent extraction, or spray drying, thus leading to the formation of spherical uniform small particles with a narrow size distribution, higher drug EE, and elimination of harmful organic solvents from the formulation. These outstanding advantages are a result of the ideal properties of SCF (especially SC-CO_2_), such as mild supercritical conditions with low critical temperature, high density, and low viscosity, and fine tunability of these properties. In addition, its environmentally friendly nature makes it a preferable choice for commercial production. Using these favorable characteristics of the SFEE process, it has been successfully applied for the development of enhanced drug delivery systems, the solubilization of poorly water-soluble drugs, physicochemical stabilization, and the solidification of liquid drugs. Based on the useful information organized and classified according to several types of drug delivery systems and active pharmaceutical ingredients, it is expected that this review will guide the evaluation of the applicability of SFEE to obtain better pharmaceutical quality when researchers in related fields are thinking about selecting a suitable manufacturing process for the preparation of desired micro/nanoparticle drug delivery systems containing their active material. In addition, it can be a good basis for examining the applicability of SFEE technology when considering the solubility, physical properties, and dosage form of the drug substance. 

As suggested above section of expert opinions, our research team is conducting a study based on the QbD approach for the SFEE process. To produce PLGA microspheres using SFEE, the following steps are being taken: (i) the critical quality attributes (CQAs) are established to prepare a high-quality formulation suitable for the quality target product profile of the sustained-release PLGA microspheres; (ii) the critical formulation and process parameters during emulsion formation, solvent removal, and particle formation are defined; (iii) critical parameters affecting the CQA selected are investigated through risk analysis; and (iv) finally, the process and formulation conditions in the design space are optimized through the design of experiments to ensure robustness and reproducibility of SFEE process. Through this QbD-based study, we will strengthen our understanding of the effect of process variables on the outcome and reduce the knowledge gap between actual and academic research in the scale-up of SFEE.

## Figures and Tables

**Figure 1 pharmaceutics-13-01928-f001:**
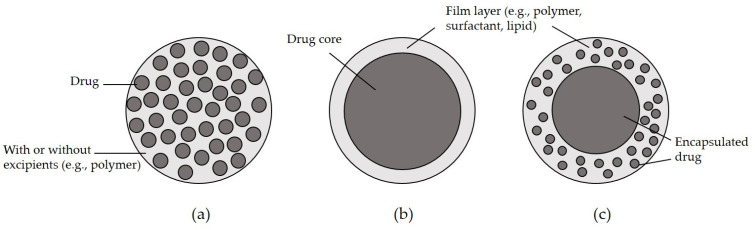
Different types of nano-/microparticle structures: (**a**) matrix, (**b**) core-shell, and (**c**) mixed type of matrix and core-shell.

**Figure 2 pharmaceutics-13-01928-f002:**
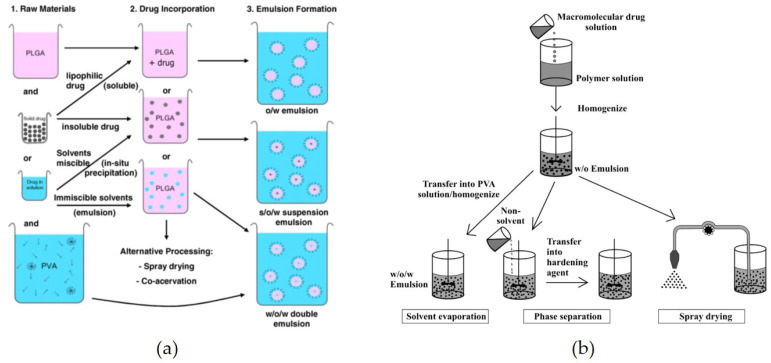
Strategies for emulsion-based micro-/nanoparticle formation: (**a**) emulsion preparation process (reprinted from [25] with permission, copyright Elsevier 2009) and (**b**) solidification process via solvent removal (reprinted from [6] with permission, copyright Elsevier 2008).

**Figure 3 pharmaceutics-13-01928-f003:**
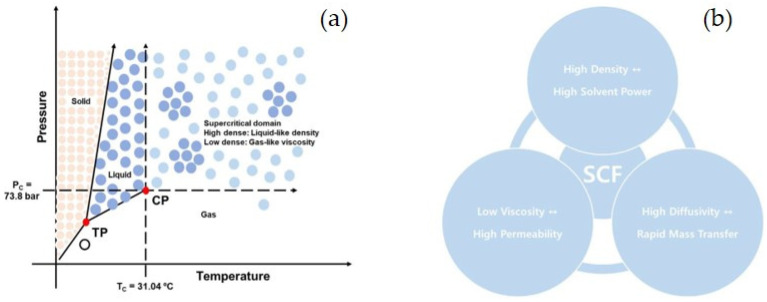
(**a**) Pressure and temperature phase diagram for CO_2_ (TC, critical temperature; TP, critical pressure; TP, triple point; and CP, critical point) and (**b**) SCF properties (reprinted from [37] with permission, copyright Elsevier 2018).

**Figure 5 pharmaceutics-13-01928-f005:**
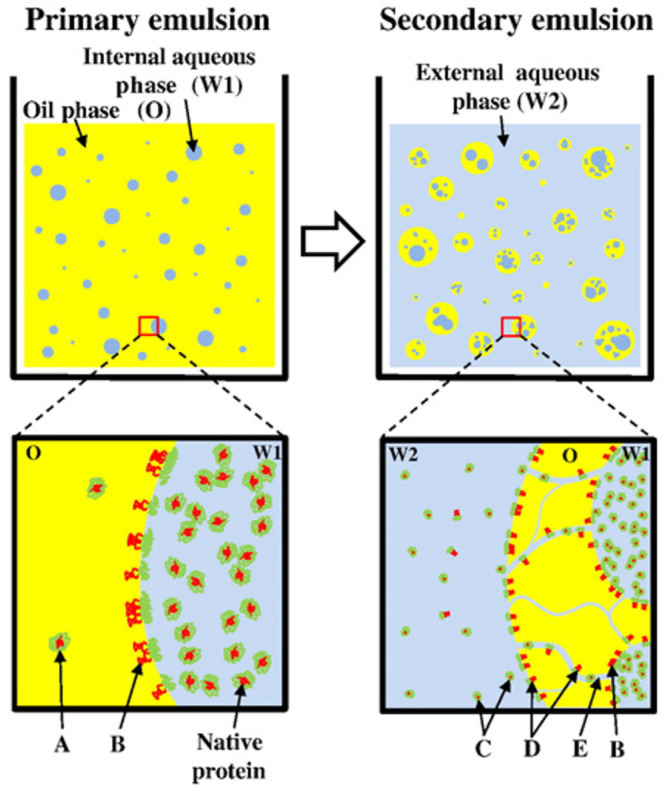
Typical *W1/O/W2* double emulsion method to prepare microspheres containing protein drug (**upper panel**) and microscopic events during the fabrication process using solvent evaporation (SE)/extraction (**lower panel**). With negligible partition of protein into oil phase (A), the organic solvent–water interface during *W1/O* emulsion results in protein denaturation (B). During generation of secondary emulsion, water channels connecting internal (W1) and external (W2) aqueous phases (E) allow proteins to escape from droplets (C) and provide more chances of protein denaturation by increased surface area of the oil–water interface (D). The water channels become pores (E) of microspheres (reprinted from [11] with permission, copyright Elsevier 2010).

**Figure 6 pharmaceutics-13-01928-f006:**
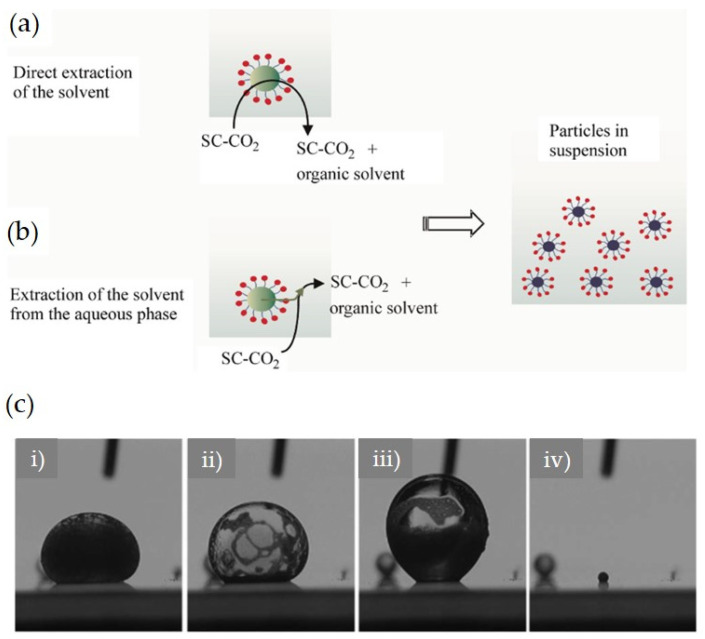
Schematic representation of the possible mass transfer pathways of the oily phase during the SFEE process. Two parallel pathways: (**a**) direct supercritical extraction upon contact between SC-CO_2_ and the organic phase into the droplet; (**b**) diffusion of the organic solvent into water followed by subsequent supercritical extraction of the solvent from the aqueous phase; and (**c**) change behavior of the volume and appearance of the organic solvent droplet as an oil phase suspended in aqueous phases overtime during the extraction process at the supercritical state: (**i**) beginning of particle formation with swelling, (**ii** and **iii**) continuous swelling and particle precipitation, and (**iv**) final condition after shrinking. Reprinted from [44] (Elsevier 2011) and [53] (Wiley 2009) with permission.

**Table 1 pharmaceutics-13-01928-t001:** Solvent removal methods for micro-/nanoparticle formation.

Method	Description	Advantages	Limitations
Solvent evaporation (SE)	- Diffusion out of the organic solvent into the outer continuous phase, and then evaporation at the interface of water and air [26]	- Available in a variety of single and double emulsions - Low facility investment cost due to the use of the already set up equipment	- Decomposition of thermolabile drugs due to high temperature use (e.g., proteins and peptides) [26] - High probability of particle formation with porous structures due to the local explosion inside the emulsion droplets [27]
Solvent extraction	- Extraction of the solvent from the dispersed phase by the continuous phase, which is optionally accompanied by SE [23]	- Available in a variety of single and double emulsions - Wide range of extraction solvents are applicable: aqueous and/or organic solvents, gases, SCF, etc.	- The need to use a large amount of extraction solvent [23,28,29] - Long process time for complete removal of organic solvents [30,31,32,33]
Spray drying (SD)	- Atomization and spraying of the starting emulsion in into a chamber with a continuous flow of hot air, and then drying	- One-step process with high production efficiency, which allows for easy scale-up [11] - Achieving particle size control and drying simultaneously through control of various process parameters - Obtaining a solid powder without the additional drying process	- Relatively high probability of particle formation with wrinkled or porous morphology undesirable for controlled release [34,35] - Unwanted change in the size of the manufactured starting emulsion droplet due to atomization - Difficulty regarding application to highly-heat-decomposable drug compounds [23].
Coacervation	- Precipitation of coating material from the continuous phase to fabricate a film layer through the formation of a coacervate phase using non-solvent or electrolytes [2,6,11]	- Applicable to both hydrophilic and hydrophobic drugs - Applicable to heat-sensitive drugs because it is performed at relatively low temperature - Production of uniform particles with a dense structure with low porosity	- Requirement of additional harmful agents for removing the outer continuous oil phase and hardening of particles - Expensive and complex process - Long processing time for complete removal of organic solvents - Frequently observed in solvent residue, and the coacervating agents are not removed in the final product [36] - Not suitable for producing particles in the nano or submicron size range [23]

**Table 2 pharmaceutics-13-01928-t002:** Pharmaceutical application cases of SFEE technology.

Purpose	Drug	Carrier	Emulsification Method	Emulsion Type	Type of SFEE	Summary	Ref.
Drug delivery system	Microencapsulation for controlled release: for hydrophobic drugs						
	Indomethacin Ketoprofen	PLGA Eudragit RS	High-speed dispersator for microparticle High-pressure homogenizer for nanoparticle	*O(EA)/W*	b-SFEE c-SFEE	- Particle size: 0.1~2 μm - Residual solvent: ≤50 ppm - The diameter of the emulsion droplet determined the size of the microsphere. - The dissolution kinetic coefficient parameter for the encapsulated drug was reduced compared to that for the non-encapsulated drug particles.	[54]
	Piroxicam	PLGA	High-speed stirrer	*O(EA)/W*	b-SFEE	- Particle size: 1~3 μm - Residual solvent: ≤40 ppm - Pressure, temperature, flow rate of SC-CO_2_, and contact time between emulsion and SC-CO_2_ affect microsphere size, distribution, and residual amount of solvent. - The higher the concentration of PLGA in the oily phase, the larger the diameter of droplets and microspheres	[24]
	Piroxicam Diclofenac Na	PLGA	High-speed stirrer Sonication+ high-speed stirrer	*O(EA)/W* *W/O(EA)/W*	b-SFEE	- Particle size: 1~3 μm - Residual solvent: ≤10 ppm - The technology using SC-CO_2_ has less residual amount of solvent, higher encapuslation efficiency, and no aggregation between particles compared to the conventional solvent evaporation (SE) technology.	[43]
	NA	PLGA	High-speed stirrer Sonication+ high-speed stirrer	*O(EA)/W* *W/O(EA)/W*	c-SFEE-PC	- Particle size: 1~3 μm - b-SFEE has a large contact area between SC-CO_2_ and the emulsion, so that the PLGA microspheres of uniform size are rapidly generated.	[44]
	Hydrocortisone	PLGA	Sonication+ high-speed Stirrer High-speed stirrer	*W/O(EA)/W* *S/O(EA)/W*	c-SFEE-PC	- Applied for formulations for injectable depot and comparison of *W/O/W* and *S/O/W* emulsions - Particle size: 1~5 μm - Residual solvent: ≤10 ppm - Excellent EE between 75% and 80% - *S/O/W* emulsion method resulted in more prolonged drug release	[55]
	Microencapsulation for controlled release: for hydrophilic drugs						
	Lysozyme	PLGA	Homogenizer	*W/O(EA)/W* *S/O(EA)/W* In situ *S/O(EA)/W*	b-SFEE	- Particle size: 0.1 ~ a few μm - Evaluation of three different emulsion types, *W/O/W*, *S/O/W*, and in situ *S/O/W*: EE results were 11%, 37%, and 49% for *W/O/W*, *S/O/W*, and in situ *S/O/W*, respectively	[25]
	Insulin	PLGA	Sonication+ high-speed homogenizer	*W/O(EA)/W*	c-SFEE-PC	- Non-collapsed spherical microparticles with size ranged 1.8~4.8 μm - Residual solvent: ≤600 ppm - EE was about 70% - Extended release for 24 days	[56,57]
	Bovine serum albumin (BSA) h-IGF	PLGA, PLA	Sonication+ high-speed Stirrer	*W/O(EA)/W*	c-SFEE-PC	- Particle size: 200~400 nm and 1~4 μm - Non-aggregated spherical micro-/nanoparticles - Very fast completion of extraction process at mild condition of 37 °C and 100 bar	[58]
	Microencapsulation for controlled release: for the combination of hydrophilic and hydrophobic drugs						
	Teriparatide and Gentamicin	PLGA/ Hydroxyapatite/ Chitosan	Sonication+ high-speed Stirrer	*O(EA)/W* *W/O(EA)/W*	c-SFEE-PC	- Application for injectable depot - Particle size: 1.4~2.2 μm - EE was up to 90% - Well-controlled drug release for 15~20 days	[59]
	NA	PLA, PCL, PLGA	Stirring and ultrasonication	*O(Acetone)/W*	c-SFEE-PC	- Particle size: 200~400 nm with polydispersity index lower to 0.1 nm - Residual solvent: ≤500 ppm - Reduced emulsion processing time and particle aggregation	[60]
	NA	Polycaprolactone (PCL)	Vortex mixing	*O(Acetone)/W*	b-SFEE	- Spontaneous emulsification method - Particle size: 200~500 nm - Preparation of monodisperse PCL nanoparticles without particle agglomeration in a short process time (only 1 h). - An increase in droplet size as polymer and surfactatnt concentration increases.	[61]
	NA	PVA	Microfluidic system	*O(EA)/W*	b-SFEE	- Particle size: 10~20 nm	[15]
	Ibuprofen	PVA/Chitosan	Microfluidic system	*O(EA)/W*	b-SFEE	- Particle size: 200~300 nm - Chitosan functionalized nanoparticles by combination with chitosan dissolution technique in acidic SC-CO_2_.	[62]
	Medroxy-progesterone acetate	PHBV	Ultrasonication	*O(DCM)/W*	b-SFEE	- Particle size: 183~850 nm - Smaller particle size of nanoparticles obtained by SFEE than that of conventional emulsion SE technology - High drug EE about 70% and low toxicity profiles - 35% drug was released within 18 h with a first order release kinetic	[63]
	Pulmonary drug delivery						
	Indomethacin Ketoprofen	Solid lipid nanoparticle (SLN)	High-pressure homogenization	*O(Chloroform)/W*	c-SFEE	- SLN suspension for pulmonary delivery via inhalation - Particle size: ~50 nm - Residual solvent: ≤20 ppm - The fraction of particles less than 3.5 μm aerodynamic particle size was greater than 90% of total aerosolized particles	[64]
	Polymeric gene delivery						
	pFlt23K, pEGFP	PLGA	Sonication	*W/O(EA)/W*	b-SFEE	- Particle size: 150~350 nm - Residual solvent: ≤50 ppm - High pDNA loading efficiency of above 98% - Effective in vitro transfection and significantly reduced VEGF secretion without cytotoxicity	[65]
	Tissue engineering						
	Growth factor	PLGA/PLA	Ultrasonication and high-speed stirring	*W/O(EA)/W*	c-SFEE-PC	- Application for injectable scaffold - Particle size: 0.4~3 μm - Drug loading property: 3~7 μg/g - Sustained release for more than 25 days - Potential as a safe biomedical device with reduced cytotoxicity with respect to microparticles obtained by conventional evaporation techniques	[66]
	Lactobacillus acidophilus	PLGA	Ultrasonication and high-speed stirring	*S/W/O(EA)/W*	b-SFEE	- Application for biodegradable spherical microdevice formed by PLGA containing bacteria trapped in a polymeric matrix - Particle size: 20 μm - Excellent EE of 80% - Shortened SFEE process time at mild condition: 30 min at 90 bar and 37 °C - Sustained-release property for longer efficacy at the implanted area	[67]
	Diglycidyl ether of bisphenol A	Poly-methyl-methacrylate (PMMA)	Ultrasonication and high-speed homogenizer	*O/O(EA)/W*	c-SFEE-PC	- Application for encapsulating structural health-monitoring agent as a form of spherical nano-/microparticles with a smooth surface - Particle size: 150~750 nm - Loading efficiency: above 55% - Enhanced EE (79%) compared to conventional SE method (18%) - Available for structural defects visualization of damaged region and promote release of self-healing material	[68]
	Nanoparticles of inorganic materials						
	TiO_2_ Nanoparticle	PLA	Sonication and high-speed stirring	*S/W(EtOH)/O(EA)/W* *S/O(EA)/W*	c-SFEE-PC	- Encapsulation for dispersion stability of nanoparticle - A stabilized colloidal nanosuspension of PLA/TiO_2_ nanostructured particles: 200~900 nm - Higher EE about 90% with uniformly dispersed TiO_2_ in PLA matrix - More homogeneous TiO_2_ dispersion in *W/O/W* based matrix particles than that of the S/O/W emulsions. - Non-reduced photocatalytic activity of TiO_2_ by the polymeric shell	[69]
	Fe_3_O_4_ magnetite	PLGA	Sonicated under magnetic stirring	*S/O(DCM)/W*	c-SFEE	- Biocompatible magnetite nanoparticle for diagnostic application as contrast agent in MRI - Particle size: 140~230 nm - Nanoparticle morphology: Janus particle type	[70]
Solubilization of poorly water-soluble drugs	Cholesterol acetate, Griseofulvin, Megestrol acetate	NA	High-pressure homogenization	*O(EA, Toluene, DCM)/W*	c-SFEE	- Application for the solubilization via particle size reduction, hence improving dissolution - Particle size: 100~1000 nm - Particle morphology: uniform prismatic crystals - Residual solvent: ≤60 ppm - Significantly improved dissolution rate (about 5 to 10 times) when compared to micronized particles	[71]
Physicochemical stabilization	Ketoprofen	PLGA	Hgh-pressure homogenization	*O(EA)/W*	b-SFEE	- Particle size: 100~200 nm - No drug nucleation during 1 week storage - Advantage of being able to create an overloaded metastable complex formulation	[72]
Solidification of liquid drug	Vitamin E	Polycaprolactone (PCL)	Vortex mixing	*O(Acetone+)/W*	b-SFEE	- Non-aggregated spherical nanoparticles with a core-shell structure - Particle size: 100~200 nm - High EE around 90% - Low residual solvent around 60 ppm	[73]

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
