# Peer review of "Pharmaceutical Applications of Supercritical Fluid Extraction of Emulsions for Micro-/Nanoparticle Formation"

_pharmaceutics, 2021, doi:10.3390/pharmaceutics13111928_

Round 1

Reviewer 1 Report

The review manuscript submitted by Park et al., is written well. However, the review mostly in the descriptive pattern like book chapter format.

The authors needs to write the different case studies with related to the selected delivery systems.

Write the current or complted clinical trails, pataents and commercial formulations through the SCF technology.

Write the expert opinon on the selected technology.

The major drawback of the technology, reproducibility of the products. How to prevent or correct the reproducibility problems with small as well as large scale production?

Line # 81: Please define EE for the first-time appearance in the manuscript.

Line # 96: Please rewrite this sentence (Difficulty in encapsulation of hydrophilic drugs with poor EE).

Line # 149: Additional references are needed.

Line # 181: Please specify thermolabile drugs (Decomposition of drugs due to high temperature use).

Line # 181: (The need to use a large amount of extraction solvent, Long process time for complete removal of organic solvents). References are needed.

Line # 194-195: A supercritical carbon dioxide (SC-CO2) is the most commonly used SCF in pharmaceutical fields due to its low critical point of temperature and pressure (Tc = 304.2 K, 195 Pc=7.38 MPa). Many other advantages need to be added.

Improve the references section and cite not more than ten years old case reports.

Author Response

Reply to Reviewer1

SUBMISSION INFORMATION:

Manuscript ID: pharmaceutics-1431538

Type of manuscript: Review

Title: Pharmaceutical Applications of Supercritical Fluid Extraction of Emulsions for Micro-/Nanoparticle Formation

Authors: Heejun Park, Jeong-Soo Kim, Sebin Kim, Eun-Sol Ha, Min-Soo Kim *, Sung-Joo Hwang *
Received: 6 October 2021

Submitted to section: Pharmaceutical Technology, Manufacturing and Devices

Submitted to Special Issue: Application of Supercritical Fluid and Compressed Gas in Pharmaceutical Technology

Dear Reviewer1,

We are grateful for the thoughtful comments of the reviewers and editorial report, whose contribution to the clarity and accuracy of the manuscript is substantial. Below is a point-by-point description of how we have revised the manuscript according to the reviewer’s comments or otherwise answered reviewer’s questions. On almost all points, we have been able to modify the manuscript exactly as suggested by the reviewers. We hope that we have addressed all points to your and reviewer’s satisfaction. In the revised manuscript, Microsoft Word's built-in track changes function was used to highlight all changes we made. Furthermore, the level of English throughout my manuscript was checked again by Editage Language Editing Services (http://editage.hibrain.net/). Editage Editing specializes in editing scientific and technical research papers written in English by authors who are not native English speakers. Please note that we have done our best to improve the level of English writing in this paper.

Please let me know if you will need anything more to process this paper. Thank you for your consideration of our manuscript.

Sincerely,

Min-Soo Kim, Ph.D.

Professor

College of Pharmacy, Pusan National University

2, Busandaehak-ro 63beon-gil, Geumjeong-gu, Busan 609-735, Republic of Korea

Tel +82 51 510 2813

Fax +82 51 513 6754

Email [email protected]

Reviewer 2 Report

The manuscript is difficult to read and understand. The explanations
given are very superficial and a reader who is not in the field will
have great difficulty in synthesizing the important information.
The descriptions which are given are very general without being able
to determine the important factors to realize this technology. Several
spelling and syntax errors are present in the document. Some abbreviations
have not been defined.

Author Response

Reply to Reviewer2

SUBMISSION INFORMATION:

Manuscript ID: pharmaceutics-1431538

Type of manuscript: Review

Title: Pharmaceutical Applications of Supercritical Fluid Extraction of Emulsions for Micro-/Nanoparticle Formation

Authors: Heejun Park, Jeong-Soo Kim, Sebin Kim, Eun-Sol Ha, Min-Soo Kim *, Sung-Joo Hwang *
Received: 6 October 2021

Submitted to section: Pharmaceutical Technology, Manufacturing and Devices

Submitted to Special Issue: Application of Supercritical Fluid and Compressed Gas in Pharmaceutical Technology

Dear Reviewer2,

We are grateful for the thoughtful comments of the reviewers and editorial report, whose contribution to the clarity and accuracy of the manuscript is substantial. Below is a point-by-point description of how we have revised the manuscript according to the reviewer’s comments or otherwise answered reviewer’s questions. On almost all points, we have been able to modify the manuscript exactly as suggested by the reviewers. We hope that we have addressed all points to your and reviewer’s satisfaction. In the revised manuscript, Microsoft Word's built-in track changes function was used to highlight all changes we made. Furthermore, the level of English throughout my manuscript was checked again by Editage Language Editing Services (http://editage.hibrain.net/). Editage Editing specializes in editing scientific and technical research papers written in English by authors who are not native English speakers. Please note that we have done our best to improve the level of English writing in this paper.

Please let me know if you will need anything more to process this paper. Thank you for your consideration of our manuscript.

Sincerely,

Min-Soo Kim, Ph.D.

Professor

College of Pharmacy, Pusan National University

2, Busandaehak-ro 63beon-gil, Geumjeong-gu, Busan 609-735, Republic of Korea

Tel +82 51 510 2813

Fax +82 51 513 6754

Email [email protected]

Reviewer #2

Q1: The manuscript is difficult to read and understand.

A1: We appreciate the reviewer’s insightful comments. I am sorry for the difficulty in reading.

English editing has been additionally performed to correct several spelling and syntax errors. In addition, we tried to revise it to be more readable and easy to understand through the below major revisions.

We have summarized the following three major problems in the structure and logical development of this review paper that made you feel like a book chapter format, and modified it as follows to improve them.

  • Our expert opinions on the application of SFEE technology and the current technology status have been sporadically commented somewhere in the middle of each chapter, but these comments are spread so widely in the main text that reviewers can be accepted as just general and superficial.

: In consideration of the opinions you suggested in Q4, expert opinion on the selected technology has been additionally described in detail in “4. Expert opinions and Perspectives”, a newly added section (P24 L741). Also, please note that this review was submitted to a special issue "Application of Supercritical Fluid and Compressed Gas in Pharmaceutical Technology" focusing on the pharmaceutical application cases of SFEE technology.

  • In previous “1. Introduction”, the process of general emulsion-based technologies for manufacturing nano-micro particles was described by dividing into emulsion preparation and solidification. The existing technologies used in each process are presented detailed. It was intended to be easy for readers to understand. However, considering that this review deals with a topic focused on the application of SFEE, it seems that the content including too detailed information about emulsion manufacturing (before the term SFEE was first mentioned in the “Introduction”) was overly structured to feel like a book chapter format. Such excessive content may be out of the scope of the subject of this review paper and do not need to be dealt with in detail.

: Thus, the summarized emulsion preparation content (previous section 1.1.1 Emulsion preparation) have been transferred in the first part describing the SFEE process (P5 L173 ~ P6 L184). Only briefly summarized emulsion-based micro-/nanoparticle preparation technology is remained in “1. Introduction” (P2 L77 ~ P2 L90). In addition, at the end of the “Introduction”, the advantages of SFEE compared to existing conventional particle formation processes were briefly described (P3 L100). Furthermore, important points that can be obtained from this review literature are introduced by adding important elements that we wanted to present through this application case-oriented review paper at the end of the introduction (P3 L105).

: In order to emphasize the advantages of the SFEE mechanism (revised section 2.2.2. Mechanism of SFEE, P7 L250), some drawbacks of conventional micro-/nanoparticle solidification processes have been newly moved to from previous section 1.1.2. to the revised section 2.2.2.1. (P7 L251).

  • The "2.2. Historical perspectives of SFEE" section was also judged to have too much content and seemed to make you feel like a book.

: The content of historical equipment development described in the previous section "2.2. Historical perspectives of SFEE" has been deleted, and only very brief introduction of the existing supercritical processes has been moved to the "2.1. Supercritical fluid (SCF)" section (P5 L143~P5 L150). The detailed history of SFEE equipment development was transferred to “2.2.1. SFEE process and apparatus” as an additional description for each equipment.

Q2: The explanations given are very superficial and a reader who is not in the field will have great difficulty in synthesizing the important information.

A2: I would like to ask my respected reviewer to please note that this review was submitted to a special issue "Application of Supercritical Fluid and Compressed Gas in Pharmaceutical Technology" focusing on the pharmaceutical application cases of SFEE technology. we have tried to improve this manuscript to be more readable and easy to understand through the above mentioned major revisions in A1. In addition, the expert opinion on the selected SFEE technology has been additionally described in detail in “4. Expert opinions and Perspectives”, a newly added section (P24 L741). Through our careful revision, we expect to overcome the superficiality of the previous version of the manuscript and deliver in-depth knowledge to our readers, so they will get great help in synthesizing the important information.

Q3: The descriptions which are given are very general without being able to determine the important factors to realize this technology.

A3: An important element that was intended to be presented through this application case-oriented review paper was added to the abstract and concluding remarks (P25 L875). In addition, to help readers understand what the important factors are in this SFEE technology, the expert opinion on the selected SFEE technology also has been additionally described in detail in “4. Expert opinions and Perspectives”, a newly added section (P24 L741).

Furthermore, in order to fundamentally explain the importance of SFEE technology in the current pharmaceutical field, the construction of main text has been reorganized to emphasize the drawbacks of convention solvent removal mechanism and the advantages of SFEE in terms of solvent and particle formation (as mentioned in A1).

Since this review paper is focusing on the application case, the important process variables that are important to the particle formation results in SFEE process are only briefly mentioned in the main text (P6 L184, P6 L195). There have been few reports on the effects of various process variables on the output. However, the lack of the fundamental understanding of the theory of particle formation in SFEE following the lack of scientific analysis of experimental data for controlling fine particle formation from lab scale to scale-up can be compensated by studies using QbD-based approaches. Thus, our research team is conducting a study based on the QbD approach for the SFEE process. Through this QbD-based study, we will strengthen our understanding of the effect of process variables on the outcome and reduce the knowledge gap between actual and academic research in the scale-up of SFEE. Please consider that after completing this further QbD based SFEE experimental research, we have a plan to write a review paper focusing on the influence of the critical process parameters (CPPs) of SFEE on responses/results based on the QbD approach. These further studies and plans have been added to concluding remarks (P26 L883).

Q4: Several spelling and syntax errors are present in the document. Some abbreviations have not been defined.

A4: I am so sorry for difficulty in reading and reviewing due to several spelling and syntax errors caused by my careless writing. English editing has been additionally performed to correct several spelling and syntax errors. Several spelling and syntax errors was corrected as shown in the revised manuscript. In addition, the definition of all used abbreviations has been added as a list according to your suggestion (P27 L895).

Round 2

Reviewer 1 Report

The manuscript modified as per the reviewer suggestions.

Reviewer 2 Report

Dear editor,

The main modifications asked in the comments sent to the authors have been taken into account. I suggest to publish the manuscript with the modifications.

Sincerely,